# Investigation on the Combined Effect of Hydroxypropyl Beta-Cyclodextrin (HPβCD) and Polysorbate in Monoclonal Antibody Formulation

**DOI:** 10.3390/ph17040528

**Published:** 2024-04-19

**Authors:** Jiayi Huang, Shiqi Hong, Lucas Yuan Hao Goh, Hailong Zhang, Tao Peng, Keat Theng Chow, Rajeev Gokhale, Vinod Tuliani

**Affiliations:** 1Pharma Applied Sciences, Roquette Asia Pacific Pte Ltd., Singapore 138588, Singapore; jiayi.huang@roquette.com (J.H.); shiqi.hong@roquette.com (S.H.); lucas.goh@roquette.com (L.Y.H.G.); hailong.zhang@roquette.com (H.Z.); keat-theng.chow@roquette.com (K.T.C.); 2Global Pharmaceutical Sciences, Roquette America Inc., 2211 Innovation Drive, Geneva, IL 60134, USA; rajeev.gokhale@roquette.com; 3Roquette Pharmaceutical Innovation Center, Lower Gwynedd Township, PA 19002, USA; vinod.tuliani@roquette.com

**Keywords:** hydroxypropyl beta-cyclodextrin, KLEPTOSE^®^ HPβCD, polysorbate, monoclonal antibody, protein formulation, excipient, protein stability, bevacizumab, ipilimumab

## Abstract

Monoclonal antibodies require careful formulation due to their inherent stability limitations. Polysorbates are commonly used to stabilize mAbs, but they are prone to degradation, which results in unwanted impurities. KLEPTOSE^®^ HPβCD (hydroxypropyl beta-cyclodextrin) has functioned as a stable stabilizer for protein formulations in our previous research. The current study investigates the collaborative impact of combining polysorbates and HPβCD as excipients in protein formulations. The introduction of HPβCD in formulations showed it considerably reduced aggregation in two model proteins, bevacizumab and ipilimumab, following exposure to various stress conditions. The diffusion interaction parameter revealed a reduction in protein–protein interactions by HPβCD. In bevacizumab formulations, the subvisible particle counts per 0.4 mL of samples in commercial formulations vs. formulations containing both HPβCD and polysorbates subjected to distinct stressors were as follows: agitation, 87,308 particles vs. 15,350 particles; light, 25,492 particles vs. 6765 particles; and heat, 1775 particles vs. 460 particles. Isothermal titration calorimetry (ITC) measurement indicated a weak interaction between PS 80 and HPβCD, with a K_D_ value of 74.7 ± 7.5 µM and binding sites of 5 × 10^–3^. Surface tension measurements illustrated that HPβCD enhanced the surface activity of polysorbates. The study suggests that combining these excipients can improve mAb stability in formulations, offering an alternative for the biopharmaceutical industry.

## 1. Introduction

Protein-based drugs, particularly monoclonal antibodies (mAbs), have revolutionized modern medicine, offering targeted treatment for a range of diseases, from autoimmune disorders to cancers. However, the inherent instability of protein molecules presents significant challenges in drug formulation and delivery [1,2]. As seen with all biotherapeutics, they are susceptible to various physical and chemical degradations such as denaturation, aggregation, and adsorption to surfaces. The long-term stability of mAbs is a key aspect in their development and requires a careful assessment of the various degradation pathways. The most common pathway of physical degradation is aggregation. Aggregates can be classified as soluble particles of submicron (<1 µm size) and insoluble particles, consisting of subvisible (<1–100 μm) and visible (>100 μm) particles [3]. Several stress factors can lead to aggregation, the most common ones during manufacturing and storage being temperature, exposure to an air–water interface, and freeze/thaw stress. Besides aggregation, chemical degradation such as deamidation, oxidation, and fragmentation can also take place, leading to significant changes in the tertiary structure of biologics. These degradation processes not only reduce the drug’s efficacy but can also induce immunogenic responses, posing risks to patient safety [4,5,6]. Excipients have been routinely used to increase the conformational stability of mAbs and to inhibit interfacial-induced aggregation.

Polysorbates are widely used in biopharmaceutical formulations to stabilize proteins based on the possible mechanism that they can compete with proteins for adsorption sites on surfaces to minimize surface-induced aggregation [7]. However, the degradation issues of polysorbates in mAbs formulations are frequently reported [8,9,10]. Their chemical diversity and complexity lead to variability in composition and performance. The degraded products such as peroxides and free fatty acids raise questions about both the lowered ability of the surfactant to protect the formulation against interfacial stresses and the impact of the degradation products on protein stability. Furthermore, these degradation products can lead to the formation of visible and subvisible particles, which are major safety and quality concerns [11].

This has led to a search and evaluation of alternative excipients in biological formulations. For example, other surfactants, carbohydrates and their derivatives, amino acid-based stabilizers, and ionic liquids have been proposed as possible alternative excipients for protein stabilization [12]. Cyclodextrins (CDs) are cyclic oligosaccharides used for the improvement of the water solubility and bioavailability of medicinal products [13,14]. β-CD and its derivatives have been approved as solubilizers and stabilizers in drug formulation by both the US and European pharmacopoeia [15]. Our previous study proved KLEPTOSE^®^ HPβCD (hydroxypropyl beta-cyclodextrin), a type of β-CD derivative, was a multifunctional and chemically stable excipient that could be applied in biopharmaceutical formulations [16,17]. Its unique molecular structure, which forms a toroid-shaped cavity, enables HPβCD to encapsulate hydrophobic molecules, thereby being able to enhance the solubility and stability of protein drugs. Unlike polysorbate, HPβCD demonstrated remarkable chemical stability under various stress conditions such as heat, light, and oxidative environments. In addition, HPβCD is widely applied as an excipient for approved parenteral applications [18]. Its weak surface activity has been well reported, and several studies have proposed its application for preventing aggregation or interfacial stress-induced particle formation. However, it was also found that HPβCD could not displace proteins from the interface as efficiently as classical surfactants. The stabilizing effect was postulated to be mostly attributed to protein−cyclodextrin interactions [19]. Yet, there is still no consensus in literature over the underlying mechanism.

While HPβCD may be an attractive alternative for polysorbates, there may be resistance for biopharmaceutical companies to switch out polysorbates completely due to decades of demonstrated efficiency and safety data [20]. This has led to a possible solution consisting of applying more than a single-component excipient, which enhances the stability and efficacy of protein drugs through synergistic interactions between multiple excipients. In traditional pharmaceutical industries, the concept of co-processing, which involves combining two or more excipients, has proven to be an advantage in stabilizing active pharmaceutical ingredients [21,22]. This concept can also be applied in biopharmaceuticals. For instance, the combination of sucrose and HPβCD or sucrose with polyvinylpyrrolidone and HPβCD was shown to be superior compared to pure sucrose formulations for long-term storage of mAbs at 40 °C [23].

Therefore, our study explored the possibility of combining polysorbates and KLEPTOSE^®^ HPβCD (HPB) in mAbs formulations to provide an alternative to the biopharmaceutical industries. In this work, two monoclonal antibodies, bevacizumab and ipilimumab of the IgG1 subclass, were used as model proteins due to their wide applications in cancer treatment. Both mAbs were administered parenterally and their commercial formulations were designed based on their specific parenteral routes. Formulations were modified with KLPETOSE^®^ HPB alone and in combination with polysorbates under various stress conditions to study the stabilization effects. The work also attempted to elucidate the stabilization mechanism of KLEPTOSE^®^ HPB as well as the collaborative or synergistic effects of KLEPTOSE^®^ HPB with polysorbates.

## 2. Results

### 2.1. Stabilizing Effects of KLEPTOSE^®^ HPB on Monoclonal Antibody

The physicochemical stability of monoclonal antibodies in formulations with KLEPTOSE^®^ HPB as a stabilizer was investigated using bevacizumab and ipilimumab antibodies as model proteins. Mechanical, heat, and light stress conditions were used to accelerate protein degradation and particle formation, and the profiles were then measured and analyzed by different analytical methods.

#### 2.1.1. Bevacizumab Stability under Various Stresses

##### Bevacizumab Aggregation and Fragmentation Profiles under Light and Heat Stresses

The stabilities of bevacizumab in 13 different formulations under different stress conditions were assessed by measuring the protein aggregation and fragmentation using SEC-HPLC. Figure 1 showed that under light stress (following the ICH Q1B guideline to treat sample under white light for 1.2 million Lux hours and UV light for 200 watt-hours per square meter) for one cycle and two cycles, increased aggregation (high-molecular-weight species) and fragmentation (low-molecular-weight species) levels were observed in all bevacizumab formulations. Consequently, reduced monomer recovery was observed in all formulations. Bevacizumab antibodies formulated in a commercial formulation buffer only (i.e., without the presence of trehalose and polysorbate 20) and a commercial formulation without polysorbate 20 were used as negative controls while bevacizumab formulated in a commercial formulation was used as positive control. Figure 1A showed that after one cycle of light stress, KLEPTOSE^®^ HPB-containing (5% *w*/*v* and 10% *w*/*v*) formulations presented significantly lower aggregation levels compared to the commercial formulation (6.05% and 5.78% vs. 7.40%). By replacing the trehalose in the commercial formulation with mannitol, even lower aggregation levels were observed for the KLEPTOSE^®^ HPB-containing (5% *w*/*v* and 10% *w*/*v*) formulations (5.59% and 5.52%). The same trend could be observed for samples being stressed for two cycles. Formulations containing 5% *w*/*v* and 10% *w*/*v* KLEPTOSE^®^ HPB with mannitol presented significantly lower aggregation levels compared to the commercial formulation (6.92% and 6.82% vs. 11.28%) (Appendix A).

At the same time, after one cycle of light stress, slightly higher fragmentation levels were observed for formulations containing trehalose and KLEPTOSE^®^ HPB compared to the commercial formulation (Figure 1B). However, the difference was not significant. By replacing trehalose with mannitol, the fragmentation levels in KLEPTOSE^®^ HPB-containing formulations were lower or not significantly different from the commercial formulation. Similar observations were also found in the samples after being stressed for two cycles (Appendix A). In terms of monomer recovery, it showed that formulations containing mannitol and 10% *w*/*v* KLEPTOSE^®^ HPB presented the highest monomer recovery of 90.58% (one cycle) and 87.38% (two cycles), while the commercial formulation showed a monomer recovery of 88.49% (one cycle) and 82.24% (two cycles) (Figure 1C and Appendix A).

Similarly, the same 13 bevacizumab formulations under heat stress at 40 °C/75% RH were monitored for four weeks. Our data showed that aggregation and fragmentation levels were increased in all formulations after two and four weeks. Figure 1D showed that after four weeks of heat stress, the aggregation levels in KLEPTOSE^®^ HPB (5% *w*/*v* and 10% *w*/*v*) with mannitol formulations were significantly lower compared to the commercial formulation (10.59% and 9.39% vs. 18.88%). The same trend can be observed for samples being stressed for two weeks only (Appendix A).

In terms of fragmentation, higher fragmentation levels were observed for all other formulations compared to the commercial formulation after four weeks of heat stress (Figure 1E). However, formulations containing KLEPTOSE^®^ HPB still presented significantly higher monomer recovery compared to the commercial formulation. The highest monomer recovery of 88.73% (two weeks) and 81.86% (four weeks) was found in the formulation containing mannitol and 10% *w*/*v* KLEPTOSE^®^ HPB, while the commercial formulation showed a significantly lower monomer recovery of 85.02% (two weeks) and 75.02% (four weeks) (Figure 1F and Appendix A).

Notably, for all the stress conditions, we observed that the combination of KLEPTOSE^®^ HPB and polysorbate provided a better stabilization effect for bevacizumab, in terms of significantly less aggregation compared to the commercial formulation.

##### Bevacizumab Charge Variant Profiles under Light Stress

Cation-exchange chromatography (CEX) was used to evaluate the charge heterogeneity of 13 different adalimumab formulations after light-stress treatment. As shown in Figure 2, when the samples were exposed to light stress according to the ICH guideline Q1B for one cycle, acid variants in all 13 formulations showed a substantial increase of approximately 20~30% compared to the unstressed samples (Appendix A), whereas the basic variants showed an increase of up to 10% after light stress (Appendix A). Meanwhile, the main peak of the commercial formulation sample suffered a significant loss of approximately 30%, whereas similar results can be observed in the trehalose and KLEPTOSE^®^ HPB-containing samples, while the main peaks in samples containing both KLEPTOSE^®^ HPB and mannitol showed lower losses (less than 25%) (Appendix A).

##### Bevacizumab Aggregation and Fragmentation Profiles under Agitation Stress

The effect of KLEPTOSE^®^ HPB on the stability of bevacizumab under agitation stress was further assessed in another five formulations. Based on our findings that mannitol was a better stabilizer than trehalose for bevacizumab, the stability of bevacizumab under agitation stress was assessed in formulations containing mannitol as an additive. Figure 2A showed that after 2 h of agitation stress, no increase in aggregation level or fragmentation was observed in all formulations. Contrarily, aggregation levels were slightly lower after agitation for all formulations (Figure 2A). This reduction in aggregate level contributed to a higher monomer recovery as observed in all formulations (Figure 2B). Compared to commercial formulation, all KLEPTOSE^®^ HPB-containing samples presented lower aggregation levels. Among all samples, the formulation containing 10% *w*/*v* KLEPTOSE^®^ HPB showed the lowest aggregation level of 2.28% and the highest monomer recovery of 97.72%. The combination of 10% KLEPTOSE^®^ HPB and PS 20 also presented a lower aggregation level compared to the commercial formulation. Although the percent difference was small (<1%), the trend remained the same in that the formulations containing KLEPTOSE^®^ HPB induced less protein aggregation. Therefore, the results showed that KLEPTOSE^®^ HPB could reduce aggregation and stabilize the mAbs in formulations, but the effectiveness varied in different scenarios.

#### 2.1.2. Ipilimumab Stability under Various Stresses

To further understand the stabilizing effects of KLEPTOSE^®^ HPB on monoclonal antibodies, another model protein, ipilimumab in seven different formulations under various stress conditions, was assessed using SEC-HPLC. As shown in Figure 3A, light stress led to significant protein aggregation in all ipilimumab formulations. After one cycle of light stress, formulations containing KLEPTOSE^®^ HPB presented significantly lower aggregation levels compared to the commercial formulation (2.16%), while the lowest aggregation levels could be found in the formulation containing 10% *w*/*v* KLEPTOSE^®^ HPB (1.13%). The same trend could be observed for samples being stressed for two cycles. Formulations containing 10% *w*/*v* KLEPTOSE^®^ HPB presented the lowest aggregation level (2.12%), while a 4.1% aggregation level was observed in the commercial formulation.

After one cycle or two cycles of light stress, only a minimal increase in fragmentation levels was observed for all formulations (Figure 3B). Formulations containing KLEPTOSE^®^ HPB presented significantly lower fragmentation levels compared to the commercial formulation. Consequently, monomer recoveries in formulation with 10% *w*/*v* KLEPTOSE^®^ HPB were highest at 98.04% and 96.95% after one cycle and two cycles of light stress, respectively (Figure 3C).

When the seven ipilimumab formulations were subjected to heat stress at 40 °C/75% RH for four weeks, only slight changes were observed in aggregation and fragmentation levels (Figure 3). Figure 3D shows that after two and four weeks of heat stress, the aggregation levels in KLEPTOSE^®^ HPB-containing formulations presented significantly lower aggregation levels compared to the commercial formulation. For fragmentation levels, as shown in Figure 3E, only approximately 0.5% of increments were observed for all samples after four weeks. Among all samples, the formulation containing 10% *w*/*v* KLEPTOSE^®^ HPB showed the lowest aggregation levels of 0.29% and 0.32%, the lowest fragmentation level of 0.83% and 1.26%, and the highest monomer recovery of 98.89% and 98.46% after two and four weeks, respectively.

Similar to the results that were observed in the bevacizumab formulations, the ipilimumab formulation combining KLEPTOSE^®^ HPB and polysorbate presented a lower aggregation level and higher monomer recovery compared to the commercial formulation under light and heat stress conditions.

Different batches of formulation studies were performed on both bevacizumab and ipilimumab, and the results showed the same trends: a. the lowest aggregation level, lowest fragmentation level, and highest monomer recovery were found in the formulation containing 10% *w*/*v* KLEPTOSE^®^ HPB; b. the formulation combining KLEPTOSE^®^ HPB and polysorbate presented a lower aggregation level and higher monomer recovery compared to the commercial formulation under stress conditions.

### 2.2. Mechanistic Studies

#### 2.2.1. Effect of KLEPTOSE^®^ HPB on Protein–Protein Interaction

To gain insight into the effect of KLEPTOSE^®^ HPB on molecular interactions, the diffusion interaction parameter, K_D_, of bevacizumab in varying concentrations of KLEPTOSE^®^ HPB was determined using a dynamic light scattering (DLS) technique. The K_D_ value is a useful indicator for the colloidal stability of a protein. Changes in protein–protein interaction (PPI) can be assessed by changes in the K_D_ values. A negative K_D_ value indicates attractive interactions while a positive K_D_ value indicates repulsive interactions [24]. Plots of bevacizumab’s mutual diffusion coefficient, Dm, measured in different KLEPTOSE^®^ HPB concentrations are shown in Figure 4A, and the corresponding K_D_ values calculated by dividing the quotient of the slope by the intercept (D_0_) of the Dm plots are illustrated in Figure 4B. At higher protein concentrations, i.e., >10 mg/mL, non-linearity of the Dm plot was observed. This deviation from linearity was expected due to thermodynamic nonidealities, crowding effects, and higher-order interactions [25]. As such, K_D_ values were determined by the linear fit of the Dm plot in the range of c = 1–10 mg/mL (Figure 4A). In Figure 4B, bevacizumab in the buffer medium has a negative K_D_ value of −20 mL/g. This is indicative of attractive protein interactions, which suggests that bevacizumab in this condition has a propensity for aggregation. With the addition of up to 100 mM KLEPTOSE^®^ HPB, the K_D_ values increased to between −5.4 and −7.7 mL/g. This suggests that the presence of KLEPTOSE^®^ HPB weakens the attractive interactions between the bevacizumab molecules. Additionally, it was noted that the largest increase in K_D_ value was observed with the addition of 25 mM or 3.5% *w*/*v* KLEPTOSE^®^ HPB concentration. Beyond this KLEPTOSE^®^ HPB concentration, K_D_ values increased marginally, indicating a limiting effect in further reducing protein–protein attractive forces for bevacizumab concentrations up to 10 mg/mL.

#### 2.2.2. Surface Activity of KLEPTOSE^®^ HPB

As introduced earlier, HPβCD is known to possess a weak surface activity and can potentially protect proteins from interfacial stress. In relation to polysorbates, the surface activity of KLEPTOSE^®^ HPB is much weaker. Static surface tension (i.e., surface tension at equilibrium) of KLEPTOSE^®^ HPB plateaued at around 60 mN/m whereas both polysorbate 20 and 80 achieved surface tension of around 35 mN/m at equilibrium (Appendix A). Hence, the protective function of KLEPTOSE^®^ HPB may not be solely attributed to its surface activity. To gain further insight into the role of KLEPTOSE^®^ HPB at the air–water interface, the dynamic surface activity of KLEPTOSE^®^ HPB and polysorbates 20 and 80 at typical usage concentrations was monitored (Figure 5A). The maximum bubble pressure (MBP) method was used to monitor the surface tension over a surface lifetime ranging from a few milliseconds to several seconds. During the agitation process, the air–water interface is said to be constantly renewed. As such, dynamic surface tension which measures the rate at which molecules migrate to the interfaces may reveal the kinetics of KLEPTOSE^®^ HPB, polysorbate, or protein at newly formed air–water interfaces. Figure 5A shows the surface tension against the surface age of different concentrations of KLEPTOSE^®^ HPB and polysorbates 20 and 80. Despite its lower surface activity, a lower initial surface tension was observed for KLEPTOSE^®^ HPB with concentrations ranging from 50 to 200 mM compared to polysorbates at 0.05% *w*/*v*. This is likely due to the much higher concentrations of KLEPTOSE^®^ HPB, allowing KLEPTOSE^®^ HPB molecules to rapidly occupy the newly formed air–water interfaces. In contrast, a slower migration kinetic from bulk to interfaces was observed for polysorbates 20 and 80 (at 0.05% *w*/*v*), possibly due to mass transfer limitation. This observation suggested that the rate at which molecules migrate to newly formed air–water interfaces is not solely surface activity-dependent but also concentration-dependent.

To understand the interplay of KLEPTOSE^®^ HPB, polysorbate, and bevacizumab at newly formed air–water interfaces, the dynamic surface tension of the bevacizumab formulations at similar concentrations used in the agitation study was measured (Figure 5B). Pure bevacizumab in the formulation buffer alone appeared to absorb or accumulate at the air–water interface as reflected by a decrease in surface tension over time. At newly formed air–water interfaces, as seen from the initial part of the graph, the bevacizumab–KLEPTOSE^®^ HPB (100 mM) formulation showed a lower surface tension than the bevacizumab–polysorbate 20 (0.04%) formulation. This suggests that KLEPTOSE^®^ HPB occupied the interface more rapidly than polysorbate 20, despite its weaker surface activity. As explained earlier, this may be due to mass-transfer limitations owing to the relatively low polysorbate 20 concentration compared to that of KLEPTOSE^®^ HPB. When used at sufficiently high concentrations, KLEPTOSE^®^ HPB may occupy the interface more rapidly than polysorbate at its typical concentrations. Nonetheless, at longer timescales, when approaching equilibrium, the surface tension of the bevacizumab-only formulation and bevacizumab–polysorbate 20 (0.04%) formulation was lower than the bevacizumab–KLEPTOSE^®^ HPB formulation, suggesting that KLEPTOSE^®^ HPB may not be as efficient in displacing bevacizumab from the interface as compared to polysorbate 20.

### 2.3. Enhancing Monoclonal Antibody Stability in Subvisible Particle Formation by the Combination of KLEPTOSE^®^ HPB and Polysorbate

As shown in Figure 1, Figure 2 and Figure 3, bevacizumab and ipilimumab formulations containing both KLEPTOSE^®^ HPB and polysorbate displayed significantly lower aggregation levels and a higher monomer recovery compared to the commercial formulation. It suggests that the combination of KLEPTOSE^®^ HPB and polysorbate can protect mAbs from degradation. Therefore, the protective effect of the combination was further evaluated by subvisible particle formation in bevacizumab formulations under various stress conditions in a testing volume of 0.4 mL per sample. Figure 6A showed that one cycle of light stress led to a significant increase in subvisible particle formation in the formulation that only contained the buffer (123,006 particles) and the commercial formulation without the presence of polysorbate 20 (89,568 particles) compared to unstressed samples (6900 and 9963 particles). With the presence of polysorbate 20 in the commercial formulation, the formation of particles was significantly reduced (25,492 particles), indicating the effectiveness of polysorbate in reducing particle formation. Compared to the commercial formulation, adding the combination of KLEPTOSE^®^ HPB (10%) and polysorbate 20 was able to further reduce the particle formation significantly (23,582 particles). Notably, replacing trehalose with mannitol was able to provide additional benefit in reducing the numbers of subvisible particles (6655 particles) for the formulation consisting of a combination of KLEPTOSE^®^ HPB (10%) and polysorbate 20.

As shown in Figure 6B, formulations under heat stress for two and four weeks did not generate higher subvisible particle counts compared to T0 samples. However, a similar trend where the combination of KLEPTOSE^®^ HPB and polysorbate 20 induced less subvisible particle formation was observed. In that case, the replacement of trehalose with mannitol did not enhance the stabilizing effect. The lowest subvisible particle numbers could be observed in formulations containing both KLEPTOSE^®^ HPB and polysorbate 20, where particle counts represented only 11.5% and 8.4% of the particles formed in the buffer-only formulation after two and four weeks of heat stress, respectively.

Furthermore, stirring for 2 h also led to a significant increase in subvisible particle formation in all formulations (shown in Figure 6C) compared to T0 samples. Compared to the formulation with buffer only, the presence of either polysorbate 20 (commercial formulation) or KLEPTOSE^®^ HPB was able to reduce the formation of particles significantly. When polysorbate was used in combination with KLEPTOSE^®^ HPB, the formulation prevented further particle formation down to 3.5% of particles formed in the formulation with buffer only. These results suggest that a combination of polysorbate 20 and KLEPTOSE^®^ HPB can synergistically modulate particle formation in formulations.

### 2.4. Interactions of KLEPTOSE^®^ HPB and Polysorbate

#### 2.4.1. Weak Binding Interactions between KLEPTOSE^®^ HPB and Polysorbate

ITC measures the heat associated with binding between two molecules, providing thermodynamic information such as the binding constant (*K_D_*), the stoichiometry (*N*), the enthalpy (∆*H_b_*), and the entropy of binding (∆*S*). Figure 7 shows that the titration of PS 80 into KLEPTOSE^®^ HPB produced corresponding exothermic peaks. According to the Gibbs free energy formula, with ∆G < 0, the reaction is spontaneous. The *K_D_* value was 74.70 ± 7.50 μM, which indicated a weak binding affinity between the two. The *N* value was 5 × 10^−3^, so about one PS 80 molecule could interact with multiple KLEPTOSE^®^ HPB molecules.

#### 2.4.2. Enhancement of Polysorbate Surface Activity by KLEPTOSE^®^ HPB

As a surfactant, polysorbates are used to stabilize air–water interfaces known to induce protein aggregation. Therefore, the high surface activity of polysorbates is important for them to compete for interfacial area and control the number of protein molecules able to aggregate. Using a tensiometer, we showed that the static surface tension of PS 80 in water reached around 39 mN/m at equilibrium. With the addition of 1%, 3%, and 5% of KLEPTOSE^®^ HPB, the static surface tension decreased to 36, 35, and 34 mN/m at equilibrium, which suggested the increment of surface activity of the solution (Figure 8A). Similarly, the static surface tension of PS 20 in water was around 36 mN/m at equilibrium, and the addition of 1%, 3%, and 5% of KLEPTOSE^®^ HPB led to a surface tension of 33, 32, and 31 mN/m at equilibrium (Figure 8B). Our results suggested that KLEPTOSE^®^ HPB was able to interact with polysorbates and enhance their surface activity, which can be beneficial for protein stabilization.

## 3. Discussion

Cyclodextrins, especially HPβCD, have been widely used as excipients in pharmaceutical products including tablets, parental solutions, nasal sprays, and eye drop solutions. While the potential of HPβCD as an excipient for protein stabilization has been a subject of investigation in numerous studies, its utilization in commercial antibody formulations has not received adequate scrutiny. In our study, we demonstrated the efficacy of KLEPTOSE^®^ HPB in significantly mitigating aggregation in two distinct antibodies, namely, bevacizumab and ipilimumab, when subjected to diverse stress conditions. Moreover, the inclusion of KLEPTOSE^®^ HPB demonstrated effectiveness in reducing particle formation within the bevacizumab formulation. In comparison to the commonly employed excipient polysorbates, KLEPTOSE^®^ HPB exhibited superior performance in protein stabilization. Interestingly, the combined use of KLEPTOSE^®^ HPB and polysorbates resulted in an unexpectedly heightened effect, prompting a comprehensive exploration of the interactions between these two excipients.

### 3.1. Mechanistic Understanding of KLEPTOSE^®^ HPB Stabilization Mechanism

In this work, KLEPTOSE^®^ HPB alone and the KLEPTOSE^®^ HPB–polysorbate combination were found to be effective in reducing protein aggregation in both bevacizumab and ipilimumab formulations under various stress conditions. It is essential to comprehend the stabilization mechanisms of KLEPTOSE^®^ HPB and the combined effects of KLEPTOSE^®^ HPB and polysorbate. Despite previous studies on the stabilization mechanism of HPβCD in mAb formulations, the existing body of research is limited, necessitating further conclusive findings. The investigation began by examining the stabilization mechanism of KLEPTOSE^®^ HPB alone. The focus was on the two most proposed stabilization mechanisms: (i) a direct interaction of HPβCD with the protein, thereby reducing the protein’s aggregation propensity, and (ii) the prevention of protein adsorption by HPβCD at the liquid/air interface, thereby reducing interfacial stress-induced protein aggregation. Using bevacizumab as the model protein, we explored the interaction of KLEPTOSE^®^ HPB with bevacizumab through dynamic light scattering (DLS) and isothermal titration calorimetry (ITC) techniques. Additionally, the maximum bubble pressure technique was employed to investigate the dynamic surface activity of bevacizumab formulations in the presence of KLEPTOSE^®^ HPB, comparing it to polysorbate 20.

The increasing K_D_ values obtained in the presence of KLEPTOSE^®^ HPB indicated that KLEPTOSE^®^ HPB weakened the attractive interaction between bevacizumab molecules (Figure 4B). Our findings corroborated well with a previously reported study by Härtl et al. [26]. KLEPTOSE^®^ HPB is postulated to change the intermolecular forces between the protein molecules, resulting in a reduction in protein–protein interaction (PPI), and hence lowering the propensity for bevacizumab aggregation. Contrary to DLS data, the ITC analysis (Appendix A) revealed no interaction between KLEPTOSE^®^ HPB and bevacizumab. The lack of interaction detected by the ITC analysis was also similarly observed in Brandenbusch et al.’s study [27]. Due to the high heat of dilution generated, which interrupted the measurement, only low concentrations of KLEPTOSE^®^ HPB (i.e., 5 mM) were used in the ITC analysis. As such, the interaction between KLEPTOSE^®^ HPB and bevacizumab might have been too weak to detect using ITC at that low concentration. Winter et al., however, managed to detect a weak interaction between HPβCD and their studied monoclonal antibodies using quartz crystal microbalance. Nonetheless, the increasing K_D_ values in the presence of KLEPTOSE^®^ HPB is evidence of the interaction between KLEPTOSE^®^ HPB and bevacizumab. The ability of KLEPTOSE^®^ HPB to reduce bevacizumab aggregation propensity can be attributed to the reduction in PPI.

Our dynamic surface tension data revealed that KLEPTOSE^®^ HPB possessed a weaker surface activity than polysorbates (Figure 5A). However, it was also noted that the rate at which molecules migrated to newly formed air–water interfaces depended on the concentration, not solely on surface activity. Bevacizumab alone was found to be highly surface active (Figure 5B) due to its amphiphilic nature, like other proteins. This means that bevacizumab will tend to adsorb onto hydrophobic interfaces like air–water interfaces. The addition of KLEPTOSE^®^ HPB to the bevacizumab formulation lowered the surface tension at the newly formed air–water interface. Despite this, at longer timescales (when approaching equilibrium), the surface tension of the bevacizumab–KLEPTOSE^®^ HPB formulation was found to be higher than the bevacizumab-only and bevacizumab–polysorbate 20 formulations. This observation suggested that KLEPTOSE^®^ HPB may not be efficient in displacing bevacizumab from the interface unlike polysorbate 20. The observation also corroborated well with another study by Serno et al., whereby their data confirmed that HPβCD did not displace the IgG from the interface [28]. Conversely, it is postulated that KLEPTOSE^®^ HPB interacts directly with bevacizumab at the air–water interface, reducing bevacizumab PPI.

### 3.2. Enhanced Effects in Protein Formulation with the Combination of KLEPTOSE^®^ HPB and Polysorbate

In our study, we observed that the addition of polysorbate in general contributed to higher aggregation and fragmentation levels in both bevacizumab and ipilimumab formulations after light stress and heat stress conditions. One possible reason might be the degradation of polysorbates by autooxidation and hydrolysis under stress conditions. The degradation might therefore lower the ability of polysorbates to protect the formulation against interfacial stresses and cause adverse impacts on the protein stability by generating degradation products [29]. In the formulations containing both KLEPTOSE^®^ HPB and polysorbates, the aggregation and fragmentation levels were lower than their respective formulations which only contained polysorbates. It suggested that the addition of KLEPTOSE^®^ HPB was able to minimize the negative effects on protein stability caused by polysorbates.

The enhancing effect of combining KLEPTOSE^®^ HPB and polysorbates can be further observed in the subvisible particle analysis of bevacizumab formulations. A major concern for a stable biologic drug formulation is that the integrity of the formulation can be significantly compromised by particle formation [30]. These visible and subvisible particles, especially in the range between ≥10 μm and ≥25 μm, have to meet the specifications of regulatory authorities [30,31]. In our study, light stress has led to significant particle formation in bevacizumab formulations without the presence of excipients. With the addition of PS 20, it is not surprising that the particle numbers dropped significantly since the ability of polysorbates to mitigate protein particle formation when exposed to liquid–air interfaces, freeze–thawing stresses, and mechanical stresses is well documented in the literature [7,12,31]. At the same time, it was noted that KLEPTOSE^®^ HPB-containing formulations prevented more particle formation as compared to PS 20 formulations, suggesting that KLEPTOSE^®^ HPB can offer better protection. Notably, the addition of KLEPTOSE^®^ HPB into the PS 20 formulation further reduced the particle numbers compared to the individual PS 20 or individual KLEPTOSE^®^ HPB formulations. A similar trend was observed under heat stress conditions while the combinative protective effect of polysorbate and KLEPTOSE^®^ HPB for reducing subvisible particle formation was more clearly observed under agitation stress. The ability of KLEPTOSE^®^ HPB to mitigate particle formation in protein formulation has been proven in our previous studies. For example, KLEPTOSE^®^ HPB was able to reduce particle formation during ultrafiltration/diafiltration of human plasma IgG and adalimumab solutions [32]. Furthermore, it was shown to suppress particle formation in adalimumab formulations under various stress conditions and the combination of PS 80 and KLEPTOSE^®^ HPB was able to reduce the particle formation efficiently under stirring conditions [16]. The possible stabilizing effect of KLEPTOSE^®^ HPB might be due to its direct interaction with the protein [33]. Polysorbate degradation in protein formulations is widely reported and the accumulation of soluble free fatty acids can trigger undesirable particles [34]. The addition of KLEPTOSE^®^ HPB might interact with polysorbates to prevent their degradation. KLEPTOSE^®^ HPB may also dissolve insoluble degradants of polysorbates due to its ability to form complexation with hydrophobic molecules [35], and hence suppress particle formation. Therefore, the combination of KLEPTOSE^®^ HPB and polysorbate was able to enhance the protein formulation.

### 3.3. Interactions between Polysorbates and KLEPTOSE^®^ HPB

To further understand the synergy between KLEPTOSE^®^ HPB and polysorbates, a set of experiments was performed. The ITC assay showed that polysorbate 80 was able to interact with KLEPTOSE^®^ HPB with a K_D_ value of 74.70 ± 7.50 μM. The ∆*H* and *T*∆*S* values were both negative, corresponding to a general predominance of van der Waals interactions, and hydrogen bonding in the KLEPTOSE^®^ HPB–polysorbate 80 interaction. It was also shown that one polysorbate 80 molecule could interact with multiple KLEPTOSE^®^ HPB molecules, suggesting that KLEPTOSE^®^ HPB might surround the polysorbate 80 in formulations to provide protection.

The stabilizing effect of polysorbates for mAbs has been extensively studied, and the most common mechanism links it to the competition between protein and surfactant at the air–water surface [28]. Therefore, the surface activity of polysorbates is important for their performance in biopharmaceutical formulations. In our study, we showed that the addition of KLEPTOSE^®^ HPB was able to increase the surface activity of both polysorbate 20 and 80 in a concentration-dependent manner. It suggested that the addition of KLEPTOSE^®^ HPB could enhance the ability of polysorbates to displace the air–water interface and hence improve protein stability.

## 4. Materials and Methods

### 4.1. Materials

The biopharma-grade mannitol and HPßCD (KLEPTOSE^®^ HPB) used in this study were from Roquette Frères (Lestrem, France). KLEPTOSE^®^ HPB has a molar substitution (M.S) of 0.62 and a molecular weight of 1387. α, α-trehalose dihydrate was purchased from Acros Organics (Fair Lawn, NJ, USA). Polysorbate 20 (PS 20) and polysorbate 80 (PS 80), sodium phosphate (monobasic, monohydrate) and sodium phosphate (dibasic, anhydrous), and diethylenetriamine pentaacetate (DTPA) were purchased from Merck KGaA (Darmstadt, Germany). All the chemicals and surfactants were multi-compendia or USP grade. Acetonitrile and methanol (HPLC grade) were purchased from J.T Baker (Phillipsburg, NJ, USA). The monoclonal antibody bevacizumab was purchased from BOC Sciences (Shirley, NY, USA). It was formulated in phosphate buffered saline (PBS) at pH 6.2 at a protein concentration of 25.1 mg/mL. Ipilimumab was produced in Chinese hamster ovary (CHO) cells and purified in-house.

### 4.2. Methods

#### 4.2.1. Excipient Preparation

Stock solutions of different excipients were prepared in Milli-Q water with different concentrations: 1% *w*/*v* PS20, 1% *w*/*v* PS80, 20% *w*/*v* KLEPTOSE^®^ HPB, 240 mg/mL trehalose, 100 mg/mL mannitol, and 1 mg/mL DTPA. All excipients were filtered using a 0.45 µm PFTE filter before usage.

#### 4.2.2. Antibody Formulation and Stability Studies

Thirteen formulations were prepared for 5 mg/mL bevacizumab as summarized in Table 1: (1) bevacizumab in formulation buffer only (50 mM sodium phosphate, pH 6.2), (2) bevacizumab in formulation buffer with 60 mg/mL trehalose, (3) bevacizumab in commercial formulation (formulation buffer with 60 mg/mL trehalose, 0.04% *w*/*v* PS 20), (4) bevacizumab in commercial formulation with 5% *w*/*v* KLEPTOSE^®^ HPB (without PS 20), (5) bevacizumab in commercial formulation with 5% *w*/*v* KLEPTOSE^®^ HPB, (6) bevacizumab in commercial formulation with 10% *w*/*v* KLEPTOSE^®^ HPB (without PS 20), (7) bevacizumab in commercial formulation with 10% *w*/*v* KLEPTOSE^®^ HPB, (8) bevacizumab in formulation buffer with 10 mg/mL mannitol, (9) bevacizumab in formulation buffer with 10 mg/mL mannitol and 0.04% *w*/*v* PS 20, (10) bevacizumab in formulation buffer with 10 mg/mL mannitol and 5% *w*/*v* KLEPTOSE^®^ HPB, (11) bevacizumab in formulation buffer with 10 mg/mL mannitol, 0.04% *w*/*v* PS 20, and 5% *w*/*v* KLEPTOSE^®^ HPB, (12) bevacizumab in formulation buffer with 10 mg/mL mannitol and 10% *w*/*v* KLEPTOSE^®^ HPB, and (13) bevacizumab in formulation buffer with 10 mg/mL mannitol, 0.04% *w*/*v* PS 20, and 10% *w*/*v* KLEPTOSE^®^ HPB. All formulations were formulated at a protein concentration of 5 mg/mL.

Seven formulations were prepared for 5 mg/mL ipilimumab as summarized in Table 2: (1) ipilimumab in formulation buffer only (5.85 mg/mL sodium chloride, 3.15 mg/mL tris hydrochloride, 0.04 mg/mL DTPA, pH 7.0), (2) ipilimumab in formulation buffer with 10 mg/mL mannitol (3) ipilimumab in commercial formulation (formulation buffer with 10 mg/mL mannitol, 0.1% *w/v* PS 80), (4) ipilimumab in commercial formulation with 5% *w/v* KLEPTOSE^®^ HPB (without PS 80), (5) ipilimumab in commercial formulation with 5% *w/v* KLEPTOSE^®^ HPB, (6) ipilimumab in commercial formulation with 10% *w/v* KLEPTOSE^®^ HPB (without PS 80), and (7) ipilimumab in commercial formulation with 10% *w/v* KLEPTOSE^®^ HPB.

All the formulations were subjected to agitation stress, heat stress, and light stress studies. For heat stress studies, according to the ICH Q1A guideline [36], all samples were subjected to a stability chamber (C500L, Weiss Technik, Reiskirchen, Germany) at 40 °C and 75% RH for 2–4 weeks. Photostability studies were performed in a photostability chamber (Pharma 500-L, Weiss Technik, Singapore, Singapore) and exposed to white light for 1.2 million Lux hours and UV light for 200 watt-hours per square meter according to the ICH Q1B guideline [37]. For agitation stress studies, all samples were stirred at 200 rpm for 2 h. Each sample was prepared in duplicate per round per study.

#### 4.2.3. Size Exclusion Chromatography (SEC)–HPLC

The stability of bevacizumab and ipilimumab in various formulations as indicated in Section 4.2.2 was evaluated using a Waters Acquity HPLC coupled with an XBridge Prem SEC column (250 Å, 2.5 μm, 7.8 × 300 mm). Samples were spun down at 21,000× *g* for 15 min at 4 °C prior to HPLC analysis. An isocratic elution was used with a mobile phase of 20 mM sodium phosphate and 150 mM sodium chloride at pH 6.8 and a flow rate of 0.4 mL/min. The absorbance was recorded at 280 nm. Peak area integration was used to determine the relative number of monomers, aggregates, and fragments. The %aggregate, %monomer, and %fragment were calculated as the area of their respective peaks divided by the total area of all peaks.

#### 4.2.4. Cation-Exchange Chromatography (CEX) for Charge Heterogeneity Profiling of Bevacizumab

A CEX–HPLC system with a Waters Protein-Pak^TM^ Hi Res WCX column (CM 7 μm, 4.6 × 100 mm) was used to analyze the charge variants of bevacizumab in various formulations as indicated in Section 4.2.2. Using a fixed pH (pH 6.1 with 25 mM of sodium phosphate) with a 0–150 mM sodium chloride gradient in 40 min, the CEX chromatographic separation of bevacizumab was achieved. Acidic variants, main isoform, and basic variants were separated and analyzed to characterize the charge variants profiles. The %acidic variants, %main isoform, and %basic variants were calculated as the area of their respective peaks divided by the total area of all peaks.

#### 4.2.5. Subvisible Particles Analysis by Micro-Flow Imaging (MFI)

MFI (MFI 5200, Protein Simple, San Jose, CA, USA) with a silane-coated 100 μm flow cell was used to evaluate the subvisible particle formation in the formulations. For each sample, 0.2 mL was used for priming the system, and 0.4 mL was analyzed. Total particle numbers were reported for >1 μm size range. The mean and standard deviation (SD) were calculated from the values of two separate measurements.

#### 4.2.6. Surface Activity Evaluation

Static surface tension was measured by a Krüss K-100 Force tensiometer (Krüss GmbH, Hamburg, Germany) using the Du Noüy ring method. A concentration series of KLEPTOSE^®^ HPB or polysorbates was automatically prepared in situ using a dosing unit. The sample volume was kept constant with the help of a second dosing unit which was used for extraction purposes. This enabled a dilution series with various concentrations to be obtained. The starting volume and concentration of each sample were fixed at 50 mL and 50 mM, respectively. Using water as a medium, a 40-dilution step was used to obtain a dilution series from 50 mM to 0.001 mM. In addition, the surface tension of the mixed samples of KLEPTOSE^®^ HPB and polysorbates described in Section 4.2.7 was evaluated using the same method, based on the concentrations of polysorbates from 1000 to 0.01 mg/mL at 25 °C.

Dynamic surface activity: a Krüss BP100 bubble-pressure tensiometer (Krüss GmbH, Hamburg, Germany) was used to determine the dynamic surface tension of KLEPTOSE^®^ HPB, polysorbates, and bevacizumab formulations. Measurements were carried out at 25 ± 0.1 °C, with the range of effective surface ages running from 10 to 50,000 ms. The diameter of the capillary used in the bubble pressure measurements was 0.249 mm.

#### 4.2.7. Binding Affinity by Isothermal Titration Calorimetry (ITC) Analysis

The binding affinity between bevacizumab and KLEPTOSE^®^ HPB and the binding affinity between KLEPTOSE^®^ HPB and PS 80 were determined by isothermal calorimetry. Calorimetric titrations were performed using a MicroCal PEAQ-ITC (Malvern, UK). Prior to each experiment, the sample cell and the syringe were rinsed with Milli-Q water. The reference cell was filled with Milli-Q water. For the measurement between bevacizumab and KLEPTOSE^®^ HPB, the sample cell was loaded with the protein solution at a concentration of 3.8 mg/mL, whereas the injection syringe was filled with KLEPTOSE^®^ HPB at a concentration of 5 mM. Both bevacizumab and KLEPTOSE^®^ HPB were formulated in 50 mM sodium phosphate, pH 6.2. For the measurement between KLEPTOSE^®^ HPB and PS 80, the sample cell was loaded with the KLEPTOSE^®^ HPB at a concentration of 200 μM, whereas the injection syringe was filled with PS 80 at a concentration of 2 mM. Both KLEPTOSE^®^ HPB and PS 80 were dissolved in Milli-Q water. The ITC instrument was equilibrated to perform both experiments at 25 °C. The initial delay time was 60 s. The reference power was set to 10 μcal/s. The titration experiment consisted of 19 injections of 4 μL with an injection speed of 0.5 μL/s. The time interval between two consecutive injections was set at 150 s to allow the heat signal to return to the baseline. During the experiments, the sample solution was continuously stirred at 500 rpm by the rotating paddle attached to the end of the syringe needle. The titration curves were analyzed using the MicroCal PEAQ-ITC Analysis software v1.41 provided with the calorimeter. A binding model with one set of sites was used to fit the data. The data of the first injection were discarded due to an inaccurate volume and concentration caused by a possible dilution of the protein solution in the syringe needle during thermal equilibration. Each experiment was repeated 3 times under the same conditions to determine the precision of the results and to ensure their reproducibility.

#### 4.2.8. Determination of Diffusion Interaction Parameter (K_D_) by Dynamic Light Scattering (DLS)

A series of bevacizumab solutions at protein concentrations ranging from 1–25 mg/mL and KLEPTOSE^®^ HPB concentrations ranging from 0–100 mM were prepared in a 50 mM phosphate buffer, pH 6.2. The hydrodynamic diameter of these samples was then measured by DLS using Zetasizer Nano ZS (Malvern Panalytical, Malvern, UK) equipped with a 633 nm helium–neon laser. The scattered light was monitored at 173° to the incident beam at a temperature of 25 °C, maintained by a Peltier controller. Prior to measurements, all solutions were filtered with a Millex-GV 0.22 μm PVDF membrane (Millipore, MA, USA) to remove any large particulate impurities. Each measurement was performed with 100 µL of filtered protein solution in a disposable plastic micro cuvette, specifically for a size measurement at 173°. Samples were first equilibrated at 25 °C for 120 s before data collection. Samples were run in triplicates using 15 acquisitions per sample. DLS data analysis was performed using ZS Xplorer software update v3.00 Microsoft Windows-based software based on cumulants. Apart from the main protein peak, a second smaller peak (belonging to KLEPTOSE^®^ HPB) at ~1–5 nm could be seen at lower protein concentrations and/or higher KLEPTOSE^®^ HPB concentrations. Hence, to calculate the diffusion coefficient (Dm) of bevacizumab, only the hydrodynamic diameter (d_H_) of the main protein peak was used, using the Stokes–Einstein equation (Equation (1)):d_H_ = KT/3πηD_m_(1)
where d_H_ is the hydrodynamic diameter (nm), K is Boltzmann’s constant, T is the absolute temperature (K), η is the viscosity of the dispersant (cP), and D_m_ is the diffusion coefficient (cm^2^/s).

The K_D_ value was then determined by a linear fit of the measured (mutual) diffusion coefficients (D_m_) as a function of protein concentration (c) (Equation (2)):D_m_ = D_0_ (1 + K_D_ · c)(2)
where D_0_ is the diffusion coefficient when concentration → 0, and K_D_ is the interaction parameter. Dividing the slope by the intercept (D_0_) results in K_D_. Due to the non-linearity at higher protein concentrations, K_D_ was determined by a linear fit in the range of c = 1–10 mg/mL.

#### 4.2.9. Statistical Analysis

Statistical significance was determined by the analysis of variance (ANOVA) in GraphPad Prism 9. A *p* value of <0.05 indicated statistical significance: * *p* < 0.05.

## 5. Conclusions

In this study, we delved into the stabilizing properties of KLEPTOSE^®^ HPB within mAb formulations, while also exploring the synergistic benefits arising from the combined use of KLEPTOSE^®^ HPB and polysorbates. A significant reduction in protein aggregation was observed when KLEPTOSE^®^ HPB was introduced into the formulations of two model proteins, bevacizumab and ipilimumab, under various stress conditions. The mechanistic study suggested the ability of KLEPTOSE^®^ HPB to reduce mAb aggregation propensity can be attributed to the reduction in PPI. Notably, the combination of KLEPTOSE^®^ HPB and polysorbates not only significantly reduced antibody aggregation but also reduced the counts of subvisible particles when subjected to agitation, light exposure, and heat stressors, as compared to commercial formulations. ITC measurements and the surface activity measurements revealed the enhanced stabilization effects might be due to the interaction between KLEPTOSE^®^ and polysorbates, while KLEPTOSE^®^ HPB was able to increase the surface activity of polysorbates. However, the mechanism of how KLEPTOSE^®^ HPB enhances the surface activity of polysorbates remains unknown. It will be interesting to conduct an in-depth evaluation of the impact of KLEPTOSE^®^ HPB on polysorbate micellar formation to elucidate the mechanism.

In summary, the findings of this study suggest that the synergistic utilization of KLEPTOSE^®^ HPB and polysorbates as excipients can enhance the stability of mAbs in formulations, providing a promising alternative for the biopharmaceutical industry to address stability challenges associated with mAb formulations. This approach holds potential for improving the quality and efficacy of biopharmaceutical products while minimizing the risks associated with polysorbate degradation. In addition, biopharma industries have shown a shift toward identifying suitable excipient combinations for various biopharmaceutical applications. By showing success in combining KLEPTOSE^®^ HPB and polysorbate, it would be promising for us to explore other excipient combinations for future biopharmaceutical applications.

## Figures and Tables

**Figure 1 pharmaceuticals-17-00528-f001:**
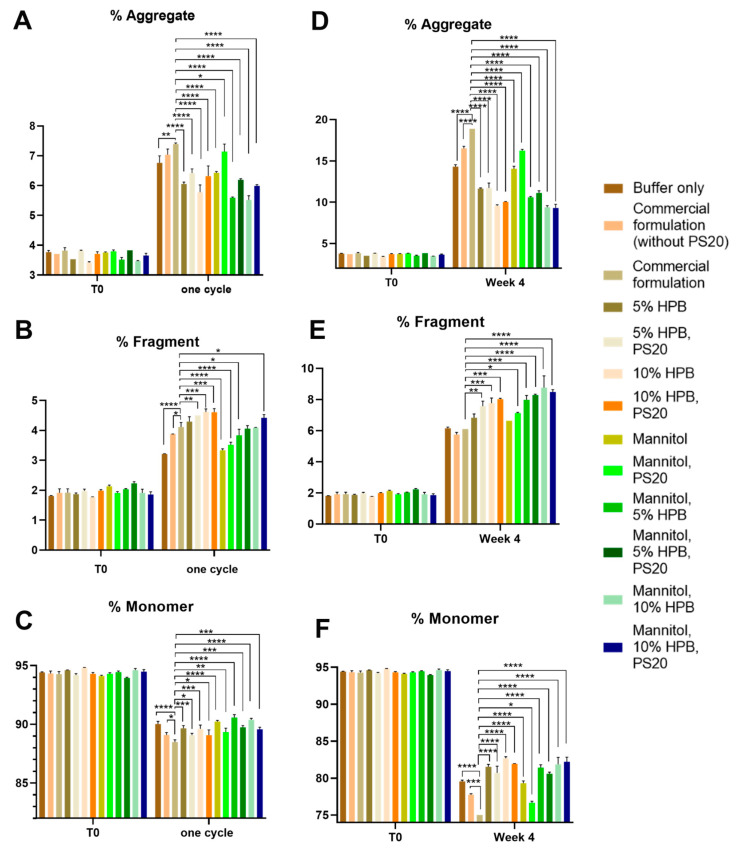
Bevacizumab size profiling under light and heat stresses. Aggregation (**A**), fragmentation (**B**), and monomer recovery (**C**) of bevacizumab in different formulations under one cycle of light stress. Aggregation (**D**), fragmentation (**E**), and monomer recovery (**F**) of bevacizumab in different formulations under heat stress for four weeks. * *p* < 0.05, ** *p* < 0.01, *** *p* < 0.001, and **** *p* < 0.0001 using one-way ANOVA compared to the commercial formulation.

**Figure 2 pharmaceuticals-17-00528-f002:**
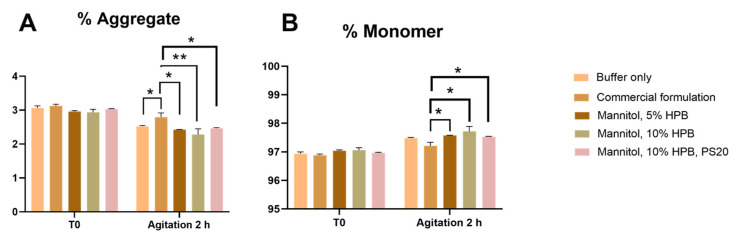
Bevacizumab size profiling under agitation stress for 2 h. Aggregation (**A**), and monomer recovery (**B**) of bevacizumab in different formulations under agitation stress. * *p* < 0.05 and ** *p* < 0.01 using one-way ANOVA compared to the commercial formulation.

**Figure 3 pharmaceuticals-17-00528-f003:**
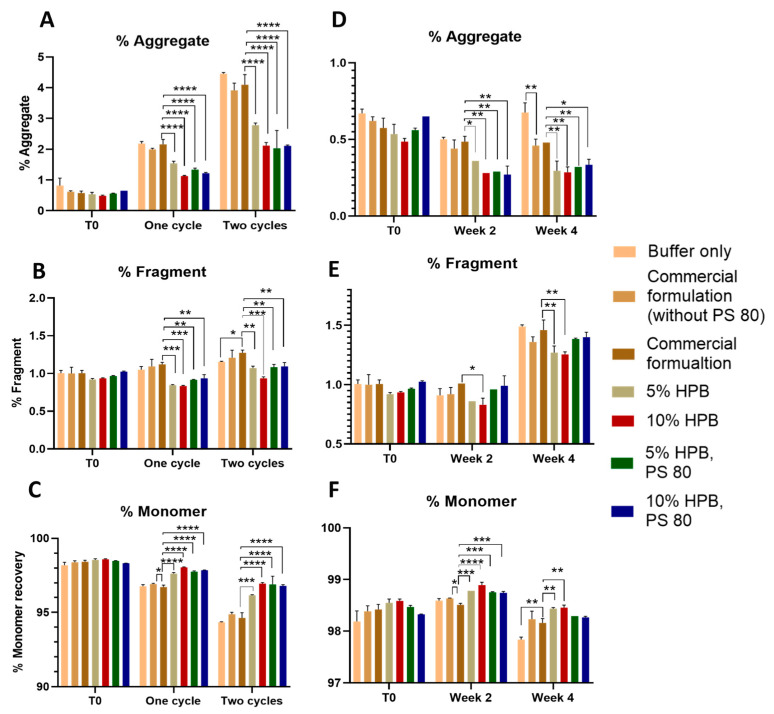
Ipilimumab size profiling under light and heat stresses. Aggregation (**A**), fragmentation (**B**), and monomer recovery (**C**) of ipilimumab in different formulations under light stress. Aggregation (**D**), fragmentation (**E**), and monomer recovery (**F**) of ipilimumab in different formulations under heat stress for two and four weeks. * *p* < 0.05, ** *p* < 0.01, *** *p* < 0.001, and **** *p* < 0.0001 using one-way ANOVA compared to the commercial formulation.

**Figure 4 pharmaceuticals-17-00528-f004:**
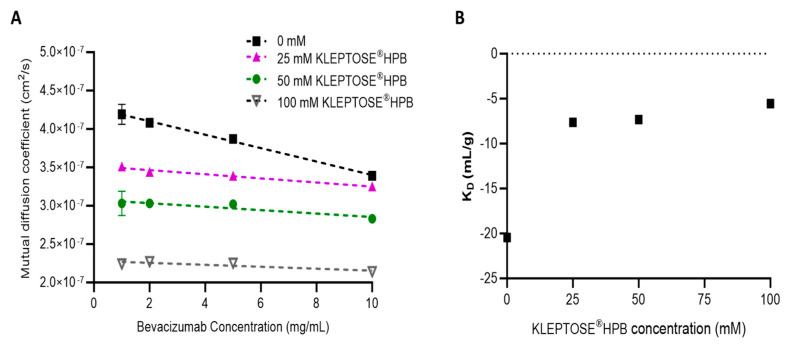
Mutual diffusion coefficient, Dm, against bevacizumab concentration plot (**A**) and K_D_ values’ plot (**B**) in the presence of different KLPETOSE^®^HPB concentrations.

**Figure 5 pharmaceuticals-17-00528-f005:**
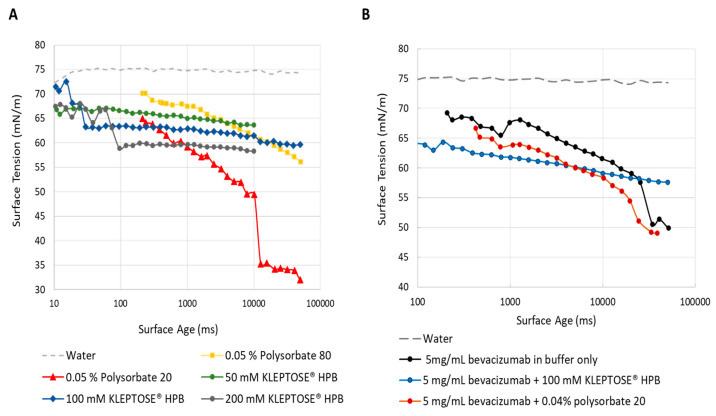
Dynamic surface tension of KLEPTOSE^®^ HPB, polysorbate 20, and polysorbate 80 (**A**) and dynamic surface tension of bevacizumab, bevacizumab with KLEPTOSE^®^ HPB, and bevacizumab with polysorbate 20 (**B**) as measured by the maximum bubble pressure technique.

**Figure 6 pharmaceuticals-17-00528-f006:**
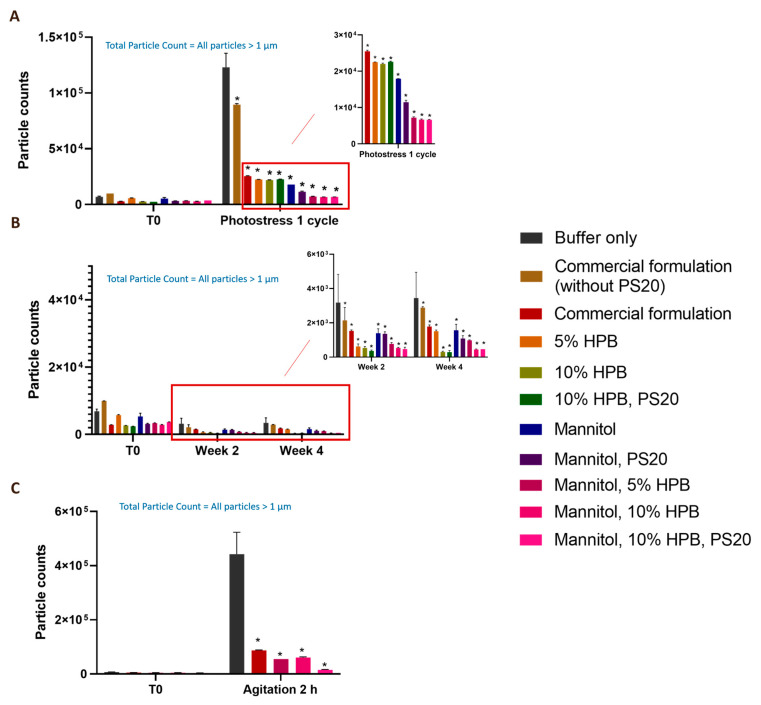
Particle formation with bevacizumab in different formulations under light (**A**), heat (**B**), and agitation stresses (**C**). * *p* < 0.05 using one-way ANOVA compared to the commercial formulation buffer.

**Figure 7 pharmaceuticals-17-00528-f007:**
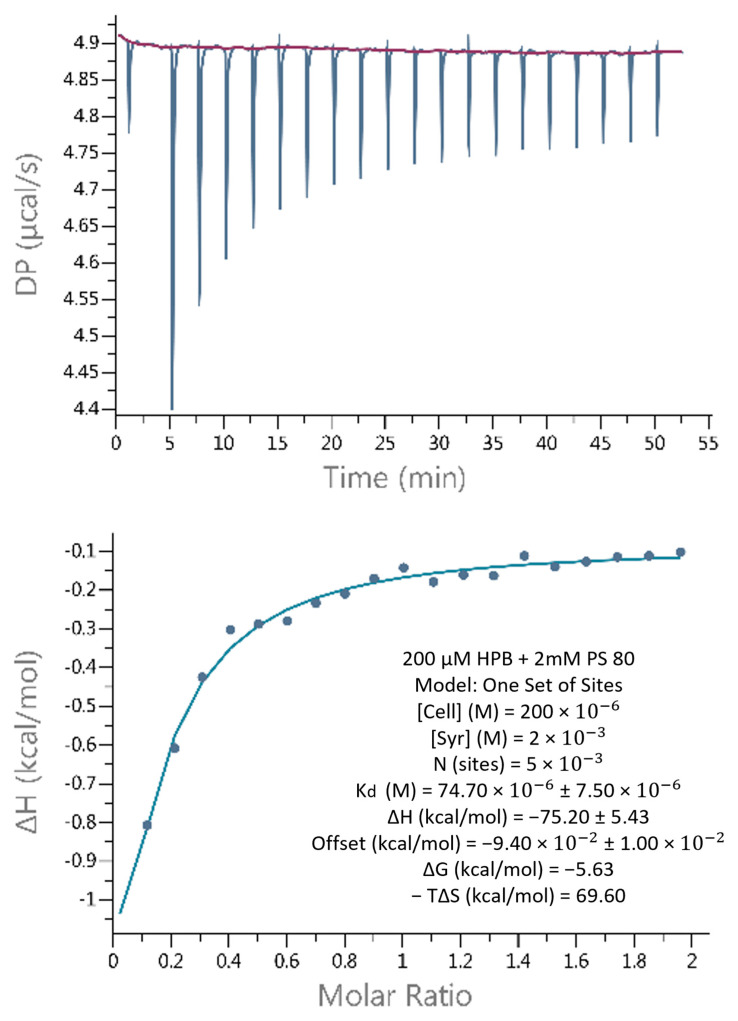
ITC data at 25 °C for titration of PS 80 into KLEPTOSE^®^ HPB. The concentration of KLEPTOSE^®^ HPB in the cell (280 μL) was 2 × 10^−6^ M, and the PS 80 concentration in the syringe was 2 × 10^−3^ M. N—reaction stoichiometry, K_d_—binding affinity, ΔH—enthalpy, ΔG—binding free energy, T—temperature, ΔS—entropy.

**Figure 8 pharmaceuticals-17-00528-f008:**
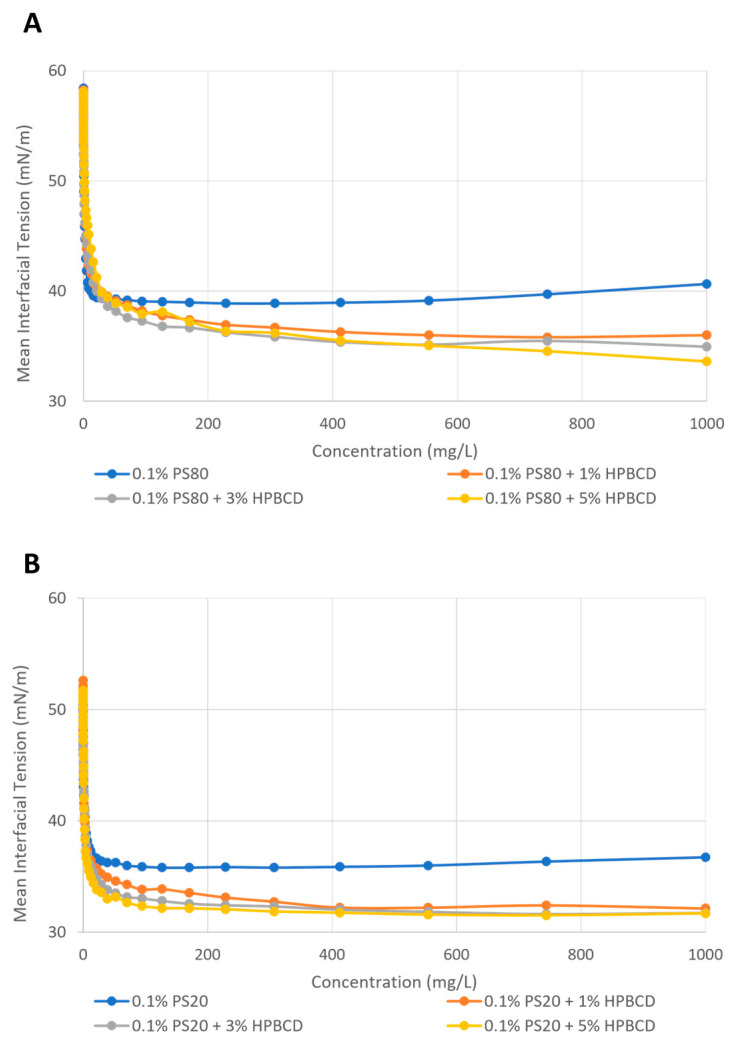
Static surface tension of 0%, 1%, 3%, and 5% of KLEPTOSE^®^ HPB with PS 80 (**A**) and PS 20 (**B**).

**Table 1 pharmaceuticals-17-00528-t001:** Formulation conditions of bevacizumab.

	Bevacizumab	Buffer	Excipients			
			Trehalose	PS 20	Mannitol	KLEPTOSE^®^ HPB
**Condition 1 (buffer only)**	5 mg/mL	50 mM sodium phosphate, pH 6.2				
**Condition 2**	5 mg/mL		60 mg/mL			
**Condition 3 (commercial formulation)**	5 mg/mL		60 mg/mL	0.04% *w*/*v*		
**Condition 4**	5 mg/mL		60 mg/mL			5% *w/v*
**Condition 5**	5 mg/mL		60 mg/mL	0.04% *w*/*v*		5% *w/v*
**Condition 6**	5 mg/mL		60 mg/mL			10% *w/v*
**Condition 7**	5 mg/mL		60 mg/mL	0.04% *w/v*		10% *w/v*
**Condition 8**	5 mg/mL				10 mg/mL	
**Condition 9**	5 mg/mL			0.04% *w/v*	10 mg/mL	
**Condition 10**	5 mg/mL				10 mg/mL	5% *w/v*
**Condition 11**	5 mg/mL			0.04% *w/v*	10 mg/mL	5% *w/v*
**Condition 12**	5 mg/mL				10 mg/mL	10% *w/v*
**Condition 13**	5 mg/mL			0.04% *w/v*	10 mg/mL	10% *w/v*

**Table 2 pharmaceuticals-17-00528-t002:** Formulation conditions of ipilimumab.

	Ipilimumab	Buffer	Excipients		
			PS 80	Mannitol	KLEPTOSE^®^ HPB
**Condition 1 (buffer only)**	5 mg/mL	5.85 mg/mL sodium chloride, 3.15 mg/mL tris hydrochloride, 0.04 mg/mL DTPA, pH 7.0			
**Condition 2**	5 mg/mL			10 mg/mL	
**Condition 3 (commercial formulation)**	5 mg/mL		0.1% *w*/*v*	10 mg/mL	
**Condition 4**	5 mg/mL				5% *w/v*
**Condition 5**	5 mg/mL		0.1% *w/v*		5% *w/v*
**Condition 6**	5 mg/mL				10% *w/v*
**Condition 7**	5 mg/mL		0.1% *w/v*		10% *w/v*

## Data Availability

The data presented in this study are available on request from the corresponding author.

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
