# Peer review of "Investigation on the Combined Effect of Hydroxypropyl Beta-Cyclodextrin (HPβCD) and Polysorbate in Monoclonal Antibody Formulation"

_pharmaceuticals, 2024, doi:10.3390/ph17040528_

Round 1
Reviewer 1 Report
Comments and Suggestions for Authors
Abstract: #87308 vs. #15350; light #25492 vs. #6765; and heat #1775 vs. #460, these numbers stands for what?
Author Response
Reviewer #1
We thank the reviewer for providing valuable feedback regarding our manuscript and below are the responses to the reviewer's comments:
- Agitation #87308 vs. #15350; light #25492 vs. #6765; and heat #1775 vs. #460, these numbers stands for what?
Response: The numbers stand for the total particle numbers generated in the formulated samples after being exposed to different stress conditions.
To clarify this issue, the sentences “In bevacizumab formulations…# 460.” in Abstract section were modified as follows (Page 1, Line 20-23):
“In bevacizumab formulations, the subvisible particle counts in per 0.4 mL of samples of commercial formulations vs. formulations containing both HPβCD and polysorbates subjected to distinct stressors were as follows: agitation, 87308 particles vs. 15350 particles; light, 25492 particles vs. 6765 particles; and heat, 1775 particles vs. 460 particles.”
Reviewer 2 Report
Comments and Suggestions for Authors
In the article " Investigation on the combined effect of hydroxypropyl beta-cy-clodextrin (HPβCD) and polysorbate in monoclonal antibody formulation", the authors explored the possibility of combining polysorbates and KLEPTOSE® HPβCD (HPB) in mAbs formulations. They analyzed the stabilizing properties of KLEPTOSE® HPB within mAb formulations, while also exploring the synergistic benefits arising from the combined use of KLEPTOSE® HPB and polysorbates. The low stability of proteins is an important problem in the development of a drug formulation that should provide adequate therapeutic efficacy. The problem addressed by the authors is part of this important area of research. However, in order for the article to proceed to further proceedings the authors should improve it. My suggestions:
Abstract
1. "In bevacizumab formulations subjected to distinct stressors, the subvisible particle counts in per 0.4 mL of samples in commercial formulations vs. formulations containing both HPβCD and polysorbates were as follows: agitation, #87308 vs. #15350; light #25492 vs. #6765; and heat #1775 vs. #460." This part of the text is unclear.
Introduction:
2. Please discuss in detail the problem of designing a drug containing a monoclonal antibody.
3. Please specify by which route the study formulation is to be administered. By the parenteral route?
4 In the text passage on the purpose of the study, it should be emphasized that bevacizumab and ipilimumab were model proteins (as indicated in line 385).
Materials and methods:
5. Please specify the purity of the reagents used.
6. Line 5050, 509, please provide literature reference for ICH guidelines.
7. Line 512, "...bevacizumab and ipilimumab in various formulations...". Which formulations are you referring to?
8. Please provide chromatographic method validation parameters: selectivity, specificity, accuracy, precision, linearity, range, limit of detection (LOD), limit of quantification (LOQ).
Conclusion:
9. In the subsection "Conclusion", please indicate the direction of further research.
10. References should be prepared in accordance with the instructions for authors.
Reviewer 3 Report
Comments and Suggestions for Authors
The work by Huang et al. extensively studies the effect of hydroxypropyl beta-cyclodextrin on formulation with PS in use of antibody drug delivery. The work is well performed in case of effect evaluation. However, I have few comments before I can recommend publication.
1) When submitting a manuscript it should be clean from author comments.
2) What are the error based on, were the experiments repeated or is it based on one batch of formulation preparation. In case it is one batch the author should repeat the experiment and show if they receive the same trend at lease for one example.
3) Is the trend of beta-cyclodextrin is specific or general for other sizes of cyclodextrin e.g. alpha and gamma because they can have similar behavior albeit they have different cavity size
4) %difference in many cases is very small. is it significant value to show such a small difference less than 1% in terms of formulation. This can vary just in different batch preparation. (I am aware that ANOVA was performed but statistics can show significant difference in insignificant result)
Author Response
We thank the reviewer for providing valuable feedback regarding our manuscript and please see the attachment for our responses to the reviewer's comments.

Round 2
Reviewer 2 Report
Comments and Suggestions for Authors
The authors have corrected the manuscript and I recommend it for further proceedings.